# Plagiocephaly after Neonatal Developmental Dysplasia of the Hip at School Age

**DOI:** 10.3390/jcm9010021

**Published:** 2019-12-19

**Authors:** A Marita Valkama, Henri I Aarnivala, Koshi Sato, Virpi Harila, Tuomo Heikkinen, Pertti Pirttiniemi

**Affiliations:** 1Department of Children and Adolescents, Oulu University Hospital, 90029 Oulu, Finland; henri.aarnivala@student.oulu.fi; 2PEDEGO Research Center, University of Oulu, 90014 Oulu, Finland; 3Medical Research Center Oulu, University of Oulu, 90014 Oulu, Finland; koshi.sato@orthodontist.jp (K.S.); virpi.harila@oulu.fi (V.H.); tuomo.heikkinen@oulu.fi (T.H.); pertti.pirttiniemi@oulu.fi (P.P.); 4Department of Oral Development and Orthodontics, Oulu University Hospital, 90014 Oulu, Finland

**Keywords:** developmental dysplasia of the hip (DDH), plagiocephaly, brachycephaly, cephalic index (CI), oblique cranial length ratio (OCLR), 2D-imaging, cross bite

## Abstract

Developmental dysplasia of the hip (DDH) may require early abduction treatment with infants sleeping on their back for the first few months of life. As sleeping on back is known to cause deformational plagiocephaly, we assessed school age children treated for dislocation or subluxation of the hip-joint in infancy. Plagiocephaly was analyzed by using cephalic index (CI) and oblique cranial length ratio (OCLR) as anthropometric measurements from 2D digital vertex view photographs. Six of the 58 (10.3%) DDH children and only one of the 62 (1.6%) control children had plagiocephaly (*p* = 0.041). Furthermore, cross bite was found in 14 (24.1%) of the DDH children and in 7 (10.3%) of the control children. Developmental dysplasia of the hip in infancy was associated with cranial asymmetries and malocclusions at school age. Preventive measures should be implemented.

## 1. Introduction

Intrauterine breech presentation and breech delivery have been shown to predispose to increased neonatal hip-joint instability requiring treatment [1,2,3]. Intrauterine constriction not only contributes to the developmental dysplasia of the hip (DDH) but also to head shape asymmetries like plagiocephaly and brachycephaly [4,5,6,7]. Furthermore DDH coexists sometimes with congenital muscular torticollis in infancy [8,9,10].

Most of the treatments for dislocation and subluxation in DDH last from 5 to 12 weeks when started early after birth [11]. Abduction splinting treatments may cause immobilization to some extend while infants lie on their back with limited possibilities for head and neck motions.

Facial asymmetry and cross bite among school children are found more common after treatments for DDH than in peers [12,13,14]. The aim of this study is to define the amount of plagiocephaly in DDH children in comparison to their unaffected peers at school age.

## 2. Subjects and Methods

### 2.1. Study Population and Study Design

The whole cohort of the treated DDH cases was 119 (0.7%) of the 16,810 newborn infants born at the University Hospital during the years 1997 to 2001. Preterm and disabled children were excluded (10 cases). The invitation letter was sent to all 109 parents by asking them to return a signed approval paper to confirm the child’s permission to participate in the study examinations and photographs. Fifty eight (53.2%) DDH children treated clinically confirmed (by neonatologists or senior pediatricians) dislocation or subluxation of the hip-joint in the University Hospital District and 62 matched control children participated in this evaluation at the mean (SD) age of 8.0 (1.4) and 7.9 (1.3) years, respectively. The control group consist of children selected from the government birth register matched by birthday, month, year, and gender according to the DDH cases. Twenty of the DDH children were male and 38 female, and 22 of the control children were male and 40 female, respectively. Treatments for DDH with abduction splinting device started soon after birth before discharging them home and the duration of treatments was 12 weeks for dislocated and 6 weeks for subluxation hips. The first ultrasound examination of the hips was taken after treatment. The study was approved by the ethics committee of the University Hospital (ET 22/2007). A written, informed consent was obtained from the parents.

### 2.2. Anthropometric Measurements for Cranial Vault Evaluation in 2D Digital Photos

For anthropometric measurements a 2D digital photograph of the vertex view was taken by the investigators with a camera fixed to the ceiling just above the child. The children were sitting in a standardized position, looking to a mirror to gain a natural head position. All the digital photographic measurements were made by an investigator, who did not know the origin group of the children nor clinical examination data. The photos were numbered and mixed. Cephalic index (CI) is the ratio between skull breadth and skull length multiplied by 100% and oblique cranial length ratio (OCLR) represents the ratio of the longer to the shorter diagonal measured at an angle of 40 degrees from the sagittal midline multiplied by 100% [15]. The cutoff point for brachycephaly was CI > 90%, and for plagiocephaly OCLR ≥ 106% with grading to severe (≥112%), moderate (≥108% and <112%), mild (≥106% and <108%), or none (100% < 106%) [15,16,17,18]. Clinical and dental examination of the children was made by a pediatrician and dentists investigators, who knew the group of the cases but did not know the existence of DP at examination. Oral dental examination and intra-oral photographs were performed, and occlusal variables including crossbite and the molar sagittal relationships were recorded.

IBM^®^ SPSS (v22.0, Armonk, New York, USA) was used for statistics. T test and cross-tabulation were used to analyze data.

## 3. Results

Term born control children had lower gestation weeks, weight, and head circumference at birth and lower present mean weight. The DDH children were more often born by cesarean section with breech presentation. Detailed characteristics of the children are in the Table 1.

One child in the DDH group was truly brachycephaly with CI 91%. Otherwise the DDH children were significantly more brachycephaly than the control children as the mean (SD) CI were 81.6 (3.7)% and 76.8 (3.5)%, respectively (*p* < 0.001). The mean (SD) OCLR values were equal with 102.5 (2.5)% in DDH and 102.4 (1.7)% in control children. Preference side in OCLR (longer right/left) was 30/28 in DDH and 49/19 in control children.

Six of the 58 (10.3%) DDH children (three mild, one moderate and two severe) and only one (moderate) of the 68 (1.5%) control children had plagiocephaly (*p* = 0.041). There were five girls in the DDH group and one boy in both groups who had plagiocephaly. Four (33.3%) of 12 children with bilateral DDH and two (11.1%) of 18 with left sided DDH had plagiocephaly. There were no remarkable association between the side of DDH and plagiocephaly.

According to the early medical data and the interview of the parents 14 (24.1%) of the DDH children have had some asymmetry in head shape, position, or neck rotation during or just after treatment in infancy. Only two of these children had plagiocephaly in the present examination. Cross bite was found in 14 (24.1%) of the DDH children and in 7 (10.3%) of the control children. Out of those three children in the DDH group were found to have plagiocephaly.

## 4. Discussion

The finding of this study is that school children treated with abduction splints during their first three months of life for DDH are more often plagiocephalic than children without DDH. Their sculls are also shorter. The right OCLR was longer in control children, but in DDH children side predominance varied. As breech presentation is highly associated to DDH it also associates to plagiocephaly and mild brachycephaly.

What is normal CI and compared to whom? We used the cut-off point of CI > 90% as lately used for teen age children [18]. There may, however, be differences between ethnic groups. Because of the nature of the retrospective study we cannot know the compression constrict in the womb before birth neither the condition of head molding or neck restriction just after birth. Early splinting with abduction devises may promote preferences for head asymmetries like brachycephaly and plagiocephaly when infants lay on their back in splints for three months. The difference of brachycephaly and plagiocephaly has lately been under discussion as it has been shown that they are very much related or a continuum of each other [7]. Also after “back to sleep” campaign normal CI is shown to increase, and therefore we cannot exactly call our DDH children brachycephalic in general but only in relation to the normal control population of the time. We do not know, either, how many of our treated or not treated children had true plagiocephaly or torticollis in infancy.

Just after birth 7.7% of healthy newborns have shown to have plagiocephaly and the amount increased to 11% when parents are guided for prevention of plagiocephaly, and to 31% without any guidance up to the age of three months [15,19]. Positional preferences or simple nursing habits after birth can cause development of deformational plagiocephaly or may provoke worsening of it [15,20]. In healthy infants plagiocephaly with the highest incidence of 20% at three to four months of age has been found to resolve to 3.3% up to the age of two years [16,21]. A recent follow-up study of children with deformational plagiocephaly in infancy reported most of the positional head deformities improve by the age of 3 to 5 years with the majority returning to normal [17]. In the present study the control children at pre pubertal age had plagiocephaly only in 1.5%. Asymmetries may still resolve after that as teenagers have been reported to have plagiocephaly only in 1.1% and brachycephaly in 1.0% [18].

Girls are shown to be more prone to neonatal DDH and boys to infant deformational plagiocephaly [1,7]. In line with this most of our plagiocephaly cases were found in girls supposedly because of over representation in DDH cases, and the only plagiocephaly case among controls was a boy. We could not find any relation between the side of unilateral DDH and ipsilateral plagiocephaly as reported earlier in infant cases [4].

Earlier it has been found that facial asymmetries by using 3-D facial imaging appear in these children who were treated with the abduction splint [12,13]. Unfortunately we could not perform the whole head 3D scanning at the time, as the method find more detailed asymmetries than analyzing 2D digital photographs.

Further follow-up is needed to find out the possible cranial asymmetries and occlusal problems in adulthood. It is also important to know, if prophylactic advises and other preventive intervention methods during early life could diminish the manifestation of DDH following it in the future.

## 5. Conclusions

Developmental dysplasia of the hip in infancy associates with cranial asymmetries like plagiocephaly and malocclusions at school age. Breech presentation at birth and abduction splints with back to sleep during early months of life may cause asymmetric head molding and obliquus scull base growth with later occlusal problems. Possibilities to prevent these should be considered early enough.

## Figures and Tables

**Table 1 jcm-09-00021-t001:** Characteristics of the children in development dysplasia of the hip (DDH) and control groups.

	DDH *N* = 58	Control *N* = 62
Gender, *n* (M/F)	20/38	22/40
Mean gestation age at birth, weeks (SD)	40.2 (1.1)	39.7 (0.9) *
Mean birth weight, g (SD)	3737.5 (437.0)	3534.7 (482.0) *
Mean birth head circumference, cm (SD)	35.7 (1.4)	34.5 (1.4) *
Mean birth length, cm (SD)	50.3 (2.2)	50.3 (1.7)
Breech presentation at birth, *n* (%)	23 (39.7)	1 (1.6) **
Caesarean section, *n* (%)	20 (34.4%)	5 (8.1) **
Mean weight, kg (SD)	31.0 (9.9)	27.4 (5.7) *
Mean height, cm (SD)	130.1 (10.4)	127.7 (9.7)
Mean head circumference, cm (SD)	54.0 (1.5)	54.3 (1.5)

* *p* < 0.05 in Independent Sample *t*-test, ** *p* < 0.01 In Chi-Square test or Fisher’s exact test.

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
