# Peer review of "Plagiocephaly after Neonatal Developmental Dysplasia of the Hip at School Age"

_jcm, 2019, doi:10.3390/jcm9010021_

Round 1

Reviewer 1 Report

I appreciate the opportunity to review this manuscript on the  “Plagiocephaly after Neonatal Developmental Dysplasia of the Hip at School Age”

The authors demonstrated in this manuscript that developmental dysplasia of the hip in infancy associates with cranial asymmetries like plagiocephaly and malocclusions at school age.

The authors state that……Fifty eight DDH children treated for dislocation or subluxation of the hip-joint in the University Hospital District and 62 matched control children participated this evaluation at the mean (SD) age of 8.0 (1.4) and 7.9 (1.3) years, respectively.

-Please clarify this statement! Were those children suffering from Developmental Dysplasia of the Hip (DDH) or Congenital Dysplasia of the Hip (CDH)?

Please clarify the abbreviations:

Developmental Dysplasia of the Hip (DDH)

Congenital Dysplasia of the Hip (CDH)

-It is confusing in the abstract and in the manuscript as the authors once talking about Developmental Dysplasia of the Hip (DDH) and then about Congenital Dysplasia of the Hip (CDH)

The authors state that…..Twenty of the DDH children were male and 38 female,and the figures for the control children were 22 and 40, respectively……

What does it mean? Please clarify!

Summery:

This manuscript is topic of interest for pediatrician as well as neurosurgeons.

Reviewer 2 Report

Although this short topical paper is of potential relevance, i still recommend the authors to further revised their paper since at present, there are still quite some uncertainties

Cases: was there a case selection, were all cases treated included in the cohort, and how is screening performed in your region since this may have major impact on the characteristics of the cases. How is the screening structured, since eg a systematic ultrasound for breech ? or driven by clinical exam, or uniformal ultrasound screening ? Are there any data on the ‘level’ of hip dysplasia at inclusion ? are there any structured advice on how to handle an infant with Pavlik treatment ? when was treatment initiated (title suggests neonatal, so first month)

The same holds true for the controls, were do they come from ? at random, hospital  population, outpatient clinics ?

Abstract: ‘Causality’ is perhaps a too strong statement, association seems better, as the authors also mention in the discussion that instructions do prevent plagiocefalie to a large extent.

Ethics: do you also need to have assent of the children in your country ?

Methods: Collection of the pictures has been described, but how have pictures been assessed. Any data on inter- or intra-rater variability, and were analyzers blinded for group allocation ? the same holds true for the oral/dental exam.

Although the 2D assessment is rather straight forward, a picture to explain what has been measured, how and when this qualified for plagiocephaly or other may be of additional value to the readership (please hereby respect privacy).

Round 2

Reviewer 1 Report

The manuscript in present form can be accepted.

Author Response

Thanks for Your helpful earlier notes and approval.

If the manuscript is accepted for publication we can use English language editong of the journal.

Reviewer 2 Report

i have read the revised version and the comments to the reviewers. I thank the authors for the revision made, but still one question: about half of the 'potential' cases were included as cases, but is there any argument for 'case selection' ? so that there are 'structural' differences between included and non-included cases ? 

second question: was the 'dental' examiner also blinded for group allocation ? this is not yet fully clear to me 

Author Response

We kindly anwered the question, please see the attachement.
